# Characterization of Porcine Urinary Bladder Matrix Hydrogels from Sodium Dodecyl Sulfate Decellularization Method

**DOI:** 10.3390/polym12123007

**Published:** 2020-12-16

**Authors:** Chen-Yu Kao, Huynh-Quang-Dieu Nguyen, Yu-Chuan Weng

**Affiliations:** 1Graduate Institute of Biomedical Engineering, National Taiwan University of Science and Technology, Taipei 10607, Taiwan; d10822805@mail.ntust.edu.tw; 2Biomedical Engineering Research Center, National Defense Medical Center, Taipei 11490, Taiwan; 3Graduate Institute of Applied Science and Technology, National Taiwan University of Science and Technology, Taipei 10607, Taiwan; 4School of Medicine, National Defense Medical Center, Taipei 11490, Taiwan; 407010125@mail.ndmctsgh.edu.tw

**Keywords:** urinary bladder matrix, decellularization, extracellular matrix hydrogel, sodium dodecyl sulfate, peracetic acid

## Abstract

Urinary bladder matrix (UBM) is one of the most studied extracellular matrixes (ECM) in the tissue engineering field. Although almost all of the UBM hydrogels were prepared by using peracetic acid (PAA), recent studies indicated that PAA was not a trustworthy way to decellularize UBM. A stronger detergent, such as sodium dodecyl sulfate (SDS), may help tackle this issue; however, its effects on the hydrogels’ characteristics remain unknown. Therefore, the objective of this study was to develop a more reliable protocol to decellularize UBM, using SDS, and to compare the characteristics of hydrogels obtained from this method to the widely employed technique, using PAA. The results indicated that SDS was superior to PAA in decellularization efficacy. Different decellularization methods led to dissimilar gelation kinetics; however, the methods did not affect other hydrogel characteristics in terms of biochemical composition, surface morphology and rheological properties. The SDS-treated hydrogels possessed excellent cytocompatibility in vitro. These results showed that the SDS decellularization method could offer a more stable and safer way to obtain acellular UBM, due to reducing immunogenicity. The hydrogels prepared from this technique had comparable characteristics as those from PAA and could be a potential candidate as a scaffold for tissue remodeling.

## 1. Introduction

The extracellular matrix (ECM) is a three-dimensional lattice structure secreted and maintained by the cells, also known as the non-cellular component of all tissues and organs [1]. ECM components include many macromolecules such as collagen, glycoproteins and glycosaminoglycan, which could not only constitute a physical scaffold for cellular adhesion, but also regulate cellular proliferation, migration or differentiation [2,3]. To extract ECM content of the tissues, many different decellularization techniques, such as physical, enzymatic, chemical or combinative methods, are employed [4].

For surgical applications, decellularized ECM have been fabricated into the form of sheets or powder. However the primary disadvantage of these two-dimensional forms is that they fail to conform to an irregularly shaped defect and cannot be applied via minimally invasive methods [5]. Therefore, injectable ECM hydrogels have emerged as an excellent candidate to tackle the problems. Not only can an ECM hydrogel be delivered by needle-based surgical techniques to fill three-dimensional spaces, but its rheological properties can also be tailored to be similar to those of the tissue of interest [6].

Among many types of ECMs, the porcine urinary bladder matrix (UBM) emerges as a unique scaffold, as it is one of few ECM scaffolds possessing an intact basement membrane, which can modulate in vitro cell growth patterns [7]. In the past few years, UBM, in the form of particulate or patches, has been broadly investigated in preclinical studies about tissue remodeling for the urethra, larynx, esophagus and heart [8]. Apart from that, UBM was also fabricated into the form of hydrogels, to study its ability for C2C12 myoblast differentiation, rat abdominal wall repair [9], axonal repair [10], neurobehavioral recovery [11], neural tissue restoration [12], neurite outgrowth [13], hippocampal neuron survival and neurite growth [14]. UBMs in all of the abovementioned research were all decellularized, using peracetic acid (PAA). However, recent studies have indicated that PAA was not an efficient and stable method to decellularize UBM, since there was still a lot of remnant DNA within the scaffold that could possibly trigger an immune response of the host to non-autologous cells [15]. In two separated studies, PAA-treated UBM had an extremely high amount of residual dsDNA with the concentration of 1990 ± 150 ng [16] and approximately 4200 ng/mg per mg dry weight [17]. Both of these remnant dsDNA amounts were too high compared to the established standards for sufficient decellularization, in which <50 ng of dsDNA per mg of dry weight is the minimal criterion that satisfies the intent of decellularization [18]. Therefore, in order to minimize the immunogenic components in UBM scaffolds, sodium dodecyl sulfate (SDS), which is a strong detergent, was used to try to tackle the issue of inadequate decellularization, with the concentration ranging from 0.1% up to 1%. However, the decellularization efficacy still varied between studies [17,19,20]. As from those results, it could be seen that obtaining adequate decellularization of UBM is not an easy task.

From another perspective, using SDS to decellularize tissues may pose a higher risk on the gelation ability of the ECM materials. In studies on the ECM hydrogel fabricated from corneas and pancreas, SDS could influence the gelation kinetics of the hydrogels, leading to failure or incomplete gel formation [2,21]. Up till now, to the best of our knowledge, there is no study using SDS to fabricate acellular UBM in the form of hydrogels. On one hand, as mentioned above, PAA was not a trustworthy method to decellularize UBM. On the other hand, SDS may have the potential of offering better decellularization efficacy, but its effects on the final hydrogel features still remain unanswered. Therefore, the objectives of this study were (1) to decellularize UBM that meets decellularization standard using SDS and (2) to compare the characteristics of hydrogels prepared from this technique with those fabricated from the widely applied protocol, using PAA in terms of decellularization efficacy, biochemical composition, gelation kinetics, mechanical strength and in vitro cytocompatibility.

## 2. Materials and Methods

### 2.1. UBM Decellularization

Bladders of 100 to 110 kg pigs were collected from Taoyuan slaughter house (Taoyuan, Taiwan). UBM which contained basement membrane and tunica propria was delaminated from the whole bladder, as previously described [8]. Three to five different UBMs were first washed with phosphate-buffered saline (PBS) and then lyophilized. The lyophilized tissues were further ground into small pieces, using a laboratory grinder. UBM pieces were stirred in 0.34 M NaCl (Sigma-Aldrich, St. Louis, MO, USA), followed by 1.28 M NaCl and PBS for 1 h each step. For the control group, which was referred to as the PAA group in this context, the tissues were decellularized with 0.1% (*v*/*v*) PAA (Ginyork, Taiwan), 4% (*v*/*v*) ethanol and 95.9% (*v*/*v*) DI water for 2 h and washed with sterilized PBS and DI water, as previously described [8]. For the testing group, which was referred to as the SDS group in this context, the tissues were decellularized with 1% (*wt*/*v*) SDS (Sigma-Aldrich, St. Louis, MO, USA) in 5× TE buffer (Sigma-Aldrich, St. Louis, MO, USA) for 24 h and then washed with DI water for an additional 48 h. The DI water was changed every 12 ± 2 h. The tissues were then incubated with 10% fetal calf serum (FCS, GE Healthcare Life Sciences, Salt Lake, UT, USA), at 37 °C, for 48 h, and then sterilized by an additional washing with 0.1% (*v*/*v*) PAA, 4% (*v*/*v*) ethanol and 95.9% (*v*/*v*) sterilized DI water for 2 h, and finally washed with sterilized PBS and DI water for 1 h each. The final decellularized tissues were then lyophilized, comminuted into fine powder and stored at −20 °C, until use.

### 2.2. Decellularization Efficiency

#### 2.2.1. dsDNA Quantification

The dsDNA was extracted from UBM powder, using a G-spin™ Total DNA Extraction Mini Kit (iNtRON Biotechnology, Seongnam, Korea). The amount of dsDNA (*n* = 3) was measured by using a Quant-iT™ PicoGreen™ dsDNA Assay Kit (Thermo-fisher scientific, Waltham, MA, USA), following manufacturer’s instruction. The fluorescence of samples (*n* = 3) was measured by using a microplate reader (Synergy H4 Hybrid, BioTek, Winooski, VT, USA) with standard fluorescein wavelengths (excitation at 480 nm, and emission at 520 nm).

#### 2.2.2. DAPI Staining

Decellularized tissues underwent a cryosection process and then were stained with 6-diamidino-2-phenylindole (DAPI, Sigma-Aldrich, St. Louis, MO, USA), to detect nuclear DNA. The fluorescent images were taken with a fluorescence microscope (CKX4, Olympus, Tokyo, Japan).

#### 2.2.3. Histological Staining

Samples were fixed with 4% paraformaldehyde (Sigma-Aldrich, St. Louis, MO, USA), dehydrated in graded alcohol and xylene, sliced into 5 μm sections and then stained with hematoxylin and eosin reagents (Merck, New York, NY, USA). The samples were visualized under an LED microscope (DM1000, Leica, Wetzlar, Germany).

#### 2.2.4. DNA Electrophoresis

The DNA fragment size of native and decellularized tissues was determined on 2% agarose gel (Bioman, New Taipei, Taiwan) containing 0.004% BioGreen safe DNA gel buffer (Bioman, New Taipei, Taiwan) and visualized with an advanced imaging system (UVP Biospectrum 500, Fisher scientific, Cambridge, UK), using a reference 100-base pair (bp) ladder (Bioman, New Taipei, Taiwan).

### 2.3. Residual SDS Detection

The remnant SDS within the ECM scaffold after decellularization was measured by using a Residual SDS detection kit (Bio Basic, Markham, ON, Canada), following manufacturer’s instructions. The samples (*n* = 3) were measured with the absorbance at 499 nm, using a plate reader (Epoch, BioTek, Winooski, VT, USA).

### 2.4. Biochemical Characterization

The amount of collagen and sGAG retention was studied, using Soluble collagen assay Sircol™ (Biocolor) and Glycosaminoglycan assay Blyscan™ (Biocolor, Carrickfergus, UK), respectively. The collagen (*n* = 4) and sGAG (*n* = 6) content of the samples were measured under 555 and 656 nm, respectively, using a microplate reader (Epoch, BioTek, Winooski, VT, USA).

### 2.5. UBM Hydrogel Preparation

UBM hydrogels were prepared as previously described [8]. Briefly, 100 of mg comminuted UBM was digested in 10 mL pepsin (Sigma-Aldrich, St. Louis, MO, USA) of 3 mg/mL in 0.01 N hydrochloric acid (HCl, Sigma-Aldrich, St. Louis, MO, USA), for 48 h, at room temperature, which was called pre-gel solution. The pre-gel samples were then neutralized to pH 7.4, using 0.1 N NaOH and 10× and 1× PBS at 4 °C, and then incubated at 37 °C for 1 h, to form hydrogels.

### 2.6. Turbidimetric Gelation Kinetics

Turbidimetric gelation kinetics were determined as previously described [8]. Then, 100 µL of neutralized pre-gel solution with a concentration of 2, 4, 6 and 8 mg/mL was transferred, per well, to a 96-well plate. The plate was immediately transferred to a microplate reader (Synergy H4 Hybrid, BioTek, Winooski, VT, USA) that was preheated to 37 °C. The absorbance at 405 nm (*n* = 4) was measured at every 3 min, for 120 min. The normalized absorbance (NA) curve was calculated by using the equation (A − A_o_)/(A_max_ − A_0_), where A was the absorbance value of specific time point; A_0_ and A_max_ were the minimum and maximum reading value, respectively; t_lag_ was defined as the time required for the gelation to start; t_1/2_ was the time required to reach 50% of the maximum absorbance; and gelation speed was calculated as the maximum slope of the growth portion of the curve.

### 2.7. Rheologic Testing

The rheological behaviors of the ECM hydrogels were determined by a modular compact rheometer (MRC 102, Anton Paar, Graz, Austria), using a 25 mm diameter parallel plate with a 0.45 µm gap. At 10 °C, the shear viscosity of the neutralized pre-gel solutions was recorded by applying a shear stress ramping from 1 to 1000 1/s. After that, the temperature was rapidly raised to 37 °C within one minute, and an oscillatory time sweep was performed, to measure the storage modulus (G’) of the hydrogels, by applying 0.5% strain at a frequency of 1 rad/s in 30 to 60 min. The final G’ was collected as the average storage modulus over the last 10 min of the time sweep test, which was the time when G’ reached a plateau without significant change. Three different batches of hydrogels were conducted (*n* = 3) for each sample.

### 2.8. Hydrogel Surface Morphology

Scanning electron microscopy (SEM) was used to examine the surface ultrastructure of ECM fibers. Hydrogels were fixed in cold 2.5% (*v*/*v*) glutaraldehyde (Sigma-Aldrich), for 24 h, at 4 °C. A series of washes in PBS and dehydrations in graded ethanol were performed. Next, the hydrogels were dried, using a critical point dryer (Samdri^®^-PVT-3D, Tousimis, MD, USA). The samples were observed under a scanning electron microscope (JSM-6500F, JEOL, Tokyo, Japan), at 10,000 magnifications. The fiber diameter (30 measurements were conduct per hydrogel) was then calculated, using Image J software (National Institutes of Health, Bethesda, MD, USA).

### 2.9. Cytocompatibility

#### 2.9.1. In Vitro Cytotoxicity

In vitro cytotoxicity of the ECM materials was evaluated according to ISO 10993-5. In brief, 10 mg of decellularized ECM powder was sterilized under UV irradiation for 4 h. Then the powder was mixed with 100 mL culture medium (DMEM containing 10% fetal bovine serum and 1% penicillin) and incubated for 1 day, at 37 °C, with 5% CO2. Media extracts (*n* = 3) were then collected, filtered to remove large particulate matters and stored at 4 °C, until further use. L929 cells (National Defense Medical Center, Taipei, Taiwan) were seeded on a 24-well plate, at a concentration of 5 × 10^3^ cells per well, and incubated at 37 °C, with 5% CO_2_, for 1 day. After that, culture medium was replaced with extracted medium. Cell viability (*n* = 3) was studied by MTT assay after 1, 2 and 3 days.

#### 2.9.2. D Surface Culture

In total, 200 µL of the neutralized pre-gel solution, at a concentration of 6 mg/mL, was injected into one well of a 24-well plate and then incubated for 1 h, to obtain complete gelation. After that, 1 mL of medium containing 1 × 10^4^ L929 cells was added on the hydrogel surface. Cell proliferation was performed by using Alamar blue assay (Bio-rad, Hercules, CA, USA) after 1, 3 and 5 days. As the control, 3 mg/mL collagen type I hydrogel from rat tail (Corning, Corning, NY, USA) was used. Cell viability on the hydrogel surface was also studied by using a LIVE/DEAD Cell Imaging Kit (488/570, Thermo Fisher Scientific, Waltham, MA, USA) following manufacture’s instruction. Cells on hydrogel surface (*n* = 3) were imaged with DMi8 microscope (Leica, Wetzlar, Germany).

#### 2.9.3. D Cell Encapsulation

First, 1 × 10^4^ L929 cells were mixed in 200 µL of neutralized pre-gel solution, at a concentration of 6 mg/mL, and then added to one well of a 24 well-plate. Then, 3 mg/mL collagen type I hydrogel from rat tail (Corning) was used as the control. The hydrogel was incubated for 1 h, at 37 °C; after that, 1 mL of medium was added. The cell-laden hydrogels were cultured for 1, 3 and 5 days and imaged (*n* = 3), using the LIVE/DEAD Cell Imaging Kit as mentioned above. The ratio of live to dead cells in each image was then calculated.

### 2.10. Statistical Analysis

All data are presented as a mean ± standard deviation. One-way ANOVA with Tuckey post hoc analyses was performed, using SPSS software (IBM Corp., Armonk, NY, USA), to determine statistical significance between experiment groups. Differences were considered to be statistically significant at *p* < 0.05.

## 3. Results

### 3.1. Decellularization Efficiency

After being prepared by physical delamination, the native UBM was then decellularized, using PAA or SDS. The results from Picogreen (Figure 1A) showed that the concentration of remnant dsDNA in the SDS-treated group (10.9 ± 2.7 ng/mg) was significantly lower than those of native UBM and PAA groups, with more than 98% of dsDNA being eliminated from the scaffolds. There was no significant difference in the cellular contents between native and PAA-treated UBM, indicating the ineffective decellularization effect of PAA. To further consolidate the results, DAPI staining assay was also performed. No intact nuclei were observed in the DAPI staining of SDS group, while there were still lots of nuclei in the tissue of native UBM and PAA-treated group (Figure 1C). A similar phenomenon was also obtained in the hematoxylin and eosin (HE) staining, in which all of the nuclei was washed out after SDS treatment (Figure 1D). In the case of dsDNA fragment size, as can be seen from Figure 1B, PAA-treated UBM possessed the DNA fragment length of more than 3000 bp, similar to that of native UBM, while there were no DNA bands detected in the SDS-treated group, indicating that the length of the remnant DNA in this decellularized UBM was smaller than 100 bp.

### 3.2. Biochemical Characterization

The soluble collagen content in the decellularized groups was significantly higher than that of native tissue, which could be explained by a loss of cellular materials in the decellularized tissues, leading to the artifact of the normalization to dry weight. However, there was no statistically difference in the collagen content between the PAA and SDS groups, with 0.52 ± 0.07 and 0.49 ± 0.01 mg soluble collagen per mg dried tissues (Figure 2A). In the case of sGAG content, nevertheless, a different pattern was observed. The sGAG level of the native tissues dropped dramatically after the decellularization process, using either PAA or SDS. The remnant sGAG inside the decellularized ECM was statistically similar between the PAA and SDS groups, with 1.49 ± 0.05 and 1.29 ± 0.01 µg sGAG per mg dried ECM (Figure 2B).

### 3.3. Turbidimetric Gelation Kinetics

The gelation kinetics of UBM hydrogels were studied by using turbidimetric analysis to define the gelation speed, lag phase and the time to reach half of the final turbidity. All of the turbidimetric gelation kinetic curves of UBM hydrogels with different concentration showed sigmoidal shapes, with hydrogel formation starting to take place after a lag period (t_lag_), and reached a stable plateau within one hour (Figure 3A,B). This observation also indicated the complete gel formation of the hydrogels. As can be seen from Figure 3C–E, in general, a lower hydrogel concentration would lead to slightly higher gelation speed and, consequently, shorter t_lag_ and t_1/2_. This phenomenon was observed in both types of UBM hydrogels obtained from PAA and SDS decellularization methods. The gelation rate, t_lag_ and t_1/2_ of PAA group were approximately 1.5 times as fast as those of the SDS group at all hydrogel concentrations (Appendix A).

### 3.4. Rheology Study

The mechanical properties of UBM hydrogels were studied by using rheology. It can be observed from Figure 4A,B that all the pre-gel solutions of both decellularization treatments possessed shear thinning properties at 10 °C. The higher the concentration would lead to higher viscosity with increased consistency factors (k). Pre-gel solutions with a lower concentration had the higher flow index (*n*), indicating the higher degree of shear thinning (Appendix A). As can be seen from Figure 4C, a higher concentration would lead to significantly stronger hydrogels. At the concentration of 4 mg/mL, the storage modulus of hydrogels from the PAA and SDS groups were only 60.8 ± 2.6 and 56.5 ± 4.9 Pa, respectively, but at the concentration of 8 mg/mL, it doubled to 145.6 ± 10.6 and 146.7 ± 7.1 Pa, respectively. However, the mechanical strength of hydrogels at the concentration of 6 mg/mL was similar to that of 8 mg/mL in both decellularization groups. It was also noticeable that the hydrogels fabricated from PAA and SDS possessed the same mechanical property at the same concentration.

### 3.5. Hydrogel Surface Morphology

SEM images qualitatively revealed that all the surface of hydrogels from both decellularization groups had a randomly oriented nano-fibrous structure (Figure 5A). The fiber size of PAA group was almost the same in all hydrogels of different concentration, with the average fiber dimeter of 0.065 µm. However, there was no consistency in terms of fiber diameter of SDS groups with average values ranging between 0.047 and 0.06 µm within ECM concentration of 2 to 8 mg/mL (Figure 5B).

### 3.6. Cytocompatibility

As can be seen from the results of MTT assay (Figure 6A), after three days of culturing in extracted medium, the cell viability of PAA and SDS groups was similar to the control, suggesting UBM powders from the PAA and SDS groups were non-toxic to the cells. After that, the UBM powder was fabricated into hydrogel, and L929 cells were cultured on the hydrogel surface. It was reported that increasing the protein concentration to obtain the desired mechanical strength of the hydrogels may affect the diffusion of oxygen, nutrients and waste across the gel [8]. Since the storage modulus of 6 mg/mL was similar to that of 8 mg/mL hydrogels, only 6 mg/mL hydrogels were chosen for 3D cell culture experiments. Moreover, 3 mg/mL collagen type I hydrogel was used as a control, as this concentration had similar collagen content as UBM hydrogels in our study. Cells proliferated significantly on the hydrogel surface after five days, according to Alamar blue assay (Figure 6B), and there was no difference among the control and testing groups. The result further verified the in vitro cytocompatibility of the hydrogels.

Both cells cultured on the surface (Figure 7A) or within the hydrogels (Figure 7B) exhibited high viability according to the live/dead assay. No live/dead ratio was reported here, as the number of dead cells was insignificant. There was no difference in the cell distribution in the surface culture experiment. However, in case of within-hydrogel culture, cells embedded in the collagen type I were dispersed more evenly throughout the entire volume, as compared to those cultured inside PAA- and SDS-treated UBM hydrogels, in which cells tended to clump together. (Optical images of cells cultured on or within hydrogels can be found in Appendix A, respectively.)

## 4. Discussion

For decades, ECM material with a variety of forms and preparation methods has been intensively studied in the field of tissue engineering and regenerative medicine [22,23]. ECM-derived materials in the form of a 2D sheet and powder are permitted to be used over the counter by the Food and Drug Administration (FDA) and served as biological implants in many clinical surgeries [6]. However, ECM material in the form of hydrogel can even broaden its potential in in vitro and in vivo application. To make ECM hydrogel, the prerequisites for preparing ECM hydrogel are that (1) the tissues must be successfully decellularized to retain the ECM scaffold and (2) this type of substratum is able to be enzymatically solubilized and neutralized into hydrogel form under appropriate physiological conditions [6]. Therefore, the first objective of this study aimed at sufficiently decellularizing UBM in order to avoid the host immune response due to the cellular remnants.

Although PAA is a well-known agent for thin tissue decellularization, such as porcine small intestinal submucosa (SIS) and UBM [6], we failed to decellularize UBM by using this method. The result was similar with recent reports indicating that PAA was not an effective reagent to decellularize UBM [16,17]. This finding was also supported by another study using PAA to decellularize SIS [24], in which the level of remaining DNA was still similar to that of native tissue, attesting to poor matrix decellularization [25]. In order to avoid this, in this study, we utilized a much stronger detergent, i.e., SDS, as the main decellularization agent, with the aid of other complementary chemicals. In terms of decellularization condition, another important aspect that must be taken into consideration is that, during decellularization, the lysis of cells releases a number of proteases into the extracellular space that can cause damage to the native ECM ultrastructure [26]. During the high concentration of 1% SDS treatment in this study, the ECM proteases might be possibly burst-released into the ECM environment. To ameliorate ECM degradation by this cation-dependent proteases, the SDS solution was prepared in 5× TE buffer, in which EDTA served as a chelator, and the pH was adjusted to 8.0, to inhibit the intrinsic proteases of the ECM scaffold. Moreover, TE buffer is also known to possess the ability to solubilize DNA or RNA, which can aid in cellular content removal [27].

The definition of effective decellularization varies greatly across published reports and is still a matter of controversy. However, in this study, we employed the considerably stringent criteria to establish sufficient decellularization, since these standards were shown to make a difference in the host response in a separate study: (1) less than 50 ng of double-stranded DNA per mg of dry weight, (2) DNA fragments less than 200 base pairs in length and (3) no visible nuclei upon histologic evaluation via HE and DAPI stains [28]. Using our decellularization protocol, we could successfully achieve all of the abovementioned strict criteria. However, the ultimate goal of decellularization is not just to get rid of the genetic material, but also to maintain structural, biochemical and biomechanical cues, and last but not least, maintain the cytocompatibility of the material. Therefore, it was imperative that residual detergent used in decellularization process and other ECM components be assessed subsequently.

Although SDS can offer sufficient cellular component removal, using SDS in decellularization protocol may compromise the cytocompatibility of the materials. It has been reported in previous studies on decellularized corneas [2], UBM [17], SIS [25] and osteochondral plugs [29] that the presence of SDS residue within ECM biomaterials has a negative impact on cellular ingrowth due to the its cellular toxicity [30]. SDS debris inside of the decellularized tissue is difficult to be disposed of, as it binds strongly to the structure of the tissues [31]. To tackle this situation, the decellularized tissues in many studies were intensively washed with DI water or PBS for at least 48 h [31]. Washing the tissues with Triton X-100 after SDS treatment was also shown to effectively lower the SDS concentration in the materials, thus enhancing the cytocompatibility of the decellularized porcine liver [32], lung [33], heart [34] and kidney [35]. In our protocol, the tissues from the SDS group were intensively washed in PBS for two days, and then washed for another two days, in FCS solution. The results in our study (Appendix A) demonstrated that the SDS was mostly eliminated from the decellularized UBM, confirming the effective SDS removal by additional washing methods.

In most soft tissues, collagen provides the essential physiological elasticity that retains the tensile strength and sGAG that provide the viscoelasticity of the material. Additionally, collagen and sGAG are two of the most crucial components which directly impact the gelation kinetics and rheological properties of ECM hydrogels [10,36]. The retention of collagen and sGAG content after decellularization was therefore studied. Similar to the PAA group, SDS did not influence the collagen content inside the decellularized UBM. This could be explained by the fact that the native collagen is generally water insoluble and could be well-preserved in the decellularized ECM scaffold [37], and SDS was also reported not to remove collagen from the tissue, in spite of being a very harsh decellularization agent [24]. However, there was a significant sGAG reduction in the decellularized ECM of both methods, as compared to the native tissue. This phenomenon was primarily due to the water-solubility characteristic of sGAG. Therefore, it is inevitable that the SGAG was rinsed out in the aqueous environment during the decellularization process.

The preparation of UBM hydrogel was first developed by Freytes et al., in 2008 [8]. However, the exact mechanism behind the gelation kinetics of ECM remains a hitherto unanswered question. It is most likely due to the complex interplay of all of the self-assembling molecules within the ECM, such as collagens, laminins and proteoglycans [8]. The results from both decellularization groups in our study demonstrated that a higher ECM concentration would lead to a lower gelation speed and longer gelation time, which was in accordance with results from the gelation kinetics analysis of heart [38], aortic adventitia [39] and SIS [40] hydrogels in other studies. Generally, the PAA group gelled faster than the SDS group, leading to the shorter t_lag_ and t_1/2_. A study by Gratzer et.al (2006) may help to explain this observation [41]. The authors found that that SDS treatment remarkably increased collagen crimp amplitude and periodicity, and increased susceptibility of collagen to degradation by the enzyme trypsin, suggesting that the treatment might have altered the native structure of collagen. This could possibly the reason why the SDS group gelled much slower than the PAA group. One noteworthy finding in our study was that, even using the harsh decellularization of SDS up to 1%, UBM hydrogels were still successfully fabricated. This is an improvement compared to other studies, in which the ECM pre-gel failed to form a gel when porcine corneas and pancreas tissue were treated with the SDS concentration above 0.1% [21,25].

In order to be applicable in minimally invasive surgery, the hydrogel must be injectable via syringes or needles. All the hydrogels in this study possessed the shear thinning characteristic, which offered great advantage during their administration, due to the reduction in resistance to when the shear strain is applied [42]. In our study, the ECM hydrogel at a concentration of 2 mg/mL, from both decellularization groups, was so weak that it could not be handled properly; therefore, it was not employed for rheology analysis. This observation was consistent with another study, in which the author found that a UBM hydrogel with a concentration of >3 mg/mL was required for the formation of hydrogel; as insufficient gelation would occur below this concentration, it could not afford to have a full rheological evaluation [43]. There was no significant difference in the storage modulus between two decellularization groups in our study. This could be explained by the similar concentration of collagen and sGAG in the decellularized tissues of PAA and SDS groups, as collagen and sGAG remarkably contribute to the mechanical properties of ECM hydrogels [10]. The results of this research were similar to other studies for UBM [8], bone [44], dermis [9] and cornea [2], reporting that a higher concentration would lead to higher viscosity and stronger mechanical strength.

Scanning electron microscopy (SEM) is often employed for visualizing the morphology of the hydrogel surface. The fiber dimeter in our study was weakly dependent on hydrogel concentration, which was consistent with UBM hydrogels from another study [9]. However, the hydrogels from PAA treatment possessed thicker fibers than those of SDS treatment in almost all concentrations. Apart from tissue of origin and protein concentration, processing methods are one of the primary factors that affect the structure of ECM materials [6]. This may be the reason for the dissimilarity in the fiber diameter between hydrogel samples in our study, as two different decellularization methods were used.

In case of materials from decellularized biological scaffolds, the cytotoxicity of the material is not only due to the immunogenicity of the material (i.e., the degree of decellularization) but also the toxicity of the residual chemicals from the decellularization reagents [45]. In our study, the extent decellularization was qualified, and there were also no SDS residues detected in the UBM scaffold, which could help preliminarily anticipate the cytocompatibility of the UBM material. According to MTT, Alamar blue and Live/dead assays, UBM powder and hydrogels from both treatments exhibited excellent cytocompatibility. There was no difference in the 3D hydrogel surface culture of L929 cells. However, cells inside the PAA or SDS groups had a tendency to clump together, compared to the even distribution of those cultured inside collagen hydrogels. This could be explained by the viscosity of the pre-gel solution during the preparation process. UBM hydrogels were prepared with a higher concentration (6 mg/mL), compared to a lower concentration, of collagen type I hydrogel (3 mg/mL). Higher viscosity may have caused the uneven mixing when the cells were added into the pre-gel solution, leading to the clumping of cells within the hydrogels. However, this did not affect the cytocompatibility of the hydrogels nor the development of the cells, as the number of dead cells was insignificant. All the results suggested that UBM hydrogels prepared from SDS decellularization method possessed an excellent cytocompatibility, as compared to the commercial available collagen type I hydrogel.

## 5. Conclusions

In summary, UBM was successfully decellularized by using SDS, and the UBM scaffold could be fabricated into the form of hydrogel. Hydrogels from SDS-treated UBM had similar characteristics compared to those prepared from the conventional decellularization method, using PAA. SDS-treated UBM hydrogels possessed lower immunogenicity and good cytocompatibility, and they could facilitate 3D culture. The results could preliminarily indicate that the UBM hydrogel prepared from our technique can be a potential candidate for tissue engineering application.

## Figures and Tables

**Figure 1 polymers-12-03007-f001:**
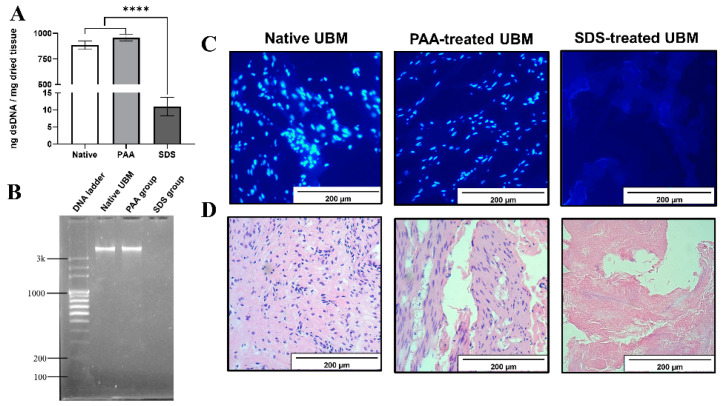
Decellularization efficiency. (**A**) dsDNA content by Picogreen assay. **** denotes a statistically significant difference (*p* < 0.0001). (**B**) DNA fragment length quantification by electrophoresis. (**C**) DAPI (6-diamidino-2-phenylindole) stain (scale bars: 200 μm). (**D**) Hematoxylin and eosin (HE) stain (scale bars: 200 μm).

**Figure 2 polymers-12-03007-f002:**
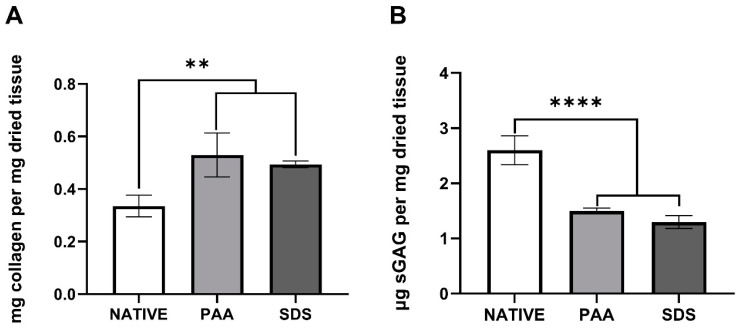
Biochemical composition of native and decellularized Urinary bladder matrix (UBM). (**A**) Collagen content (mg/mg) (*n* = 4). (**B**) sGAG content (µg/mg) (*n* = 6). ** and **** denote a statistically significant difference with *p* < 0.005 and 0.0001, respectively.

**Figure 3 polymers-12-03007-f003:**
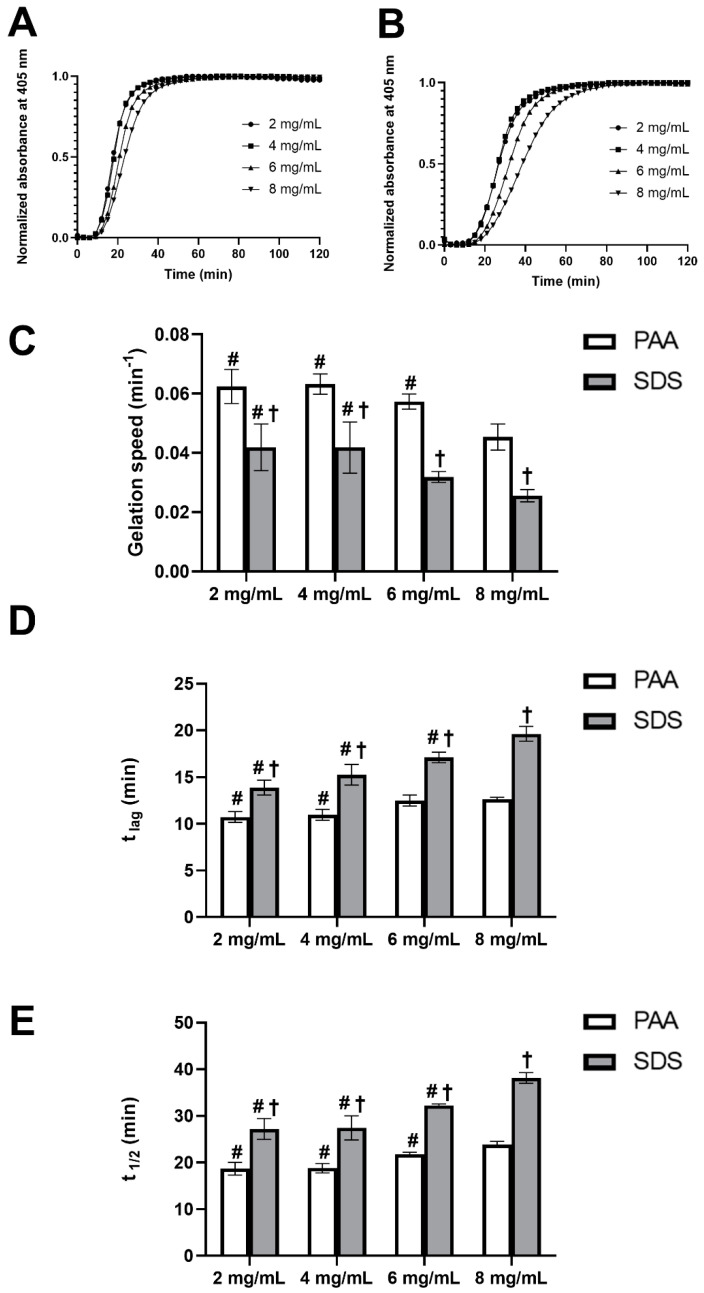
Turbidimetric gelation kinetics of UBM hydrogels. Normalized representative curves of UBM hydrogels from peracetic acid (PAA) (**A**) and sodium dodecyl sulfate (SDS) (**B**) decellularization methods. Speed (**C**), t lag (**D**) and t 1/2 (**E**) of UBM hydrogels. # denotes statistically difference from the 8 mg/mL concentration of the same decellularization group type, and † denotes statistically difference between PAA and SDS group at the same concentration (*n* = 4) (*p* < 0.05).

**Figure 4 polymers-12-03007-f004:**
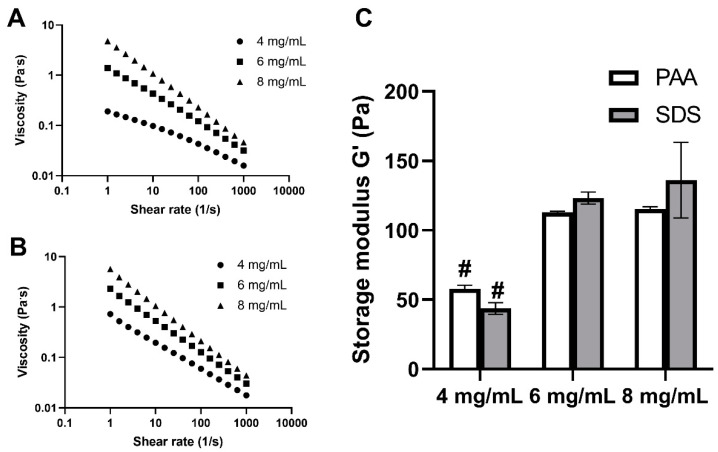
Rheological properties of UBM hydrogels. Viscosity of pre-gel solutions of PAA group (**A**) and SDS group (**B**). Storage modulus (G’) of hydrogels from PAA- and SDS-treated tissues (**C**). # denotes statistically difference from the 8 mg/mL concentration of the same decellularization group (*n* = 3) (*p* < 0.05).

**Figure 5 polymers-12-03007-f005:**
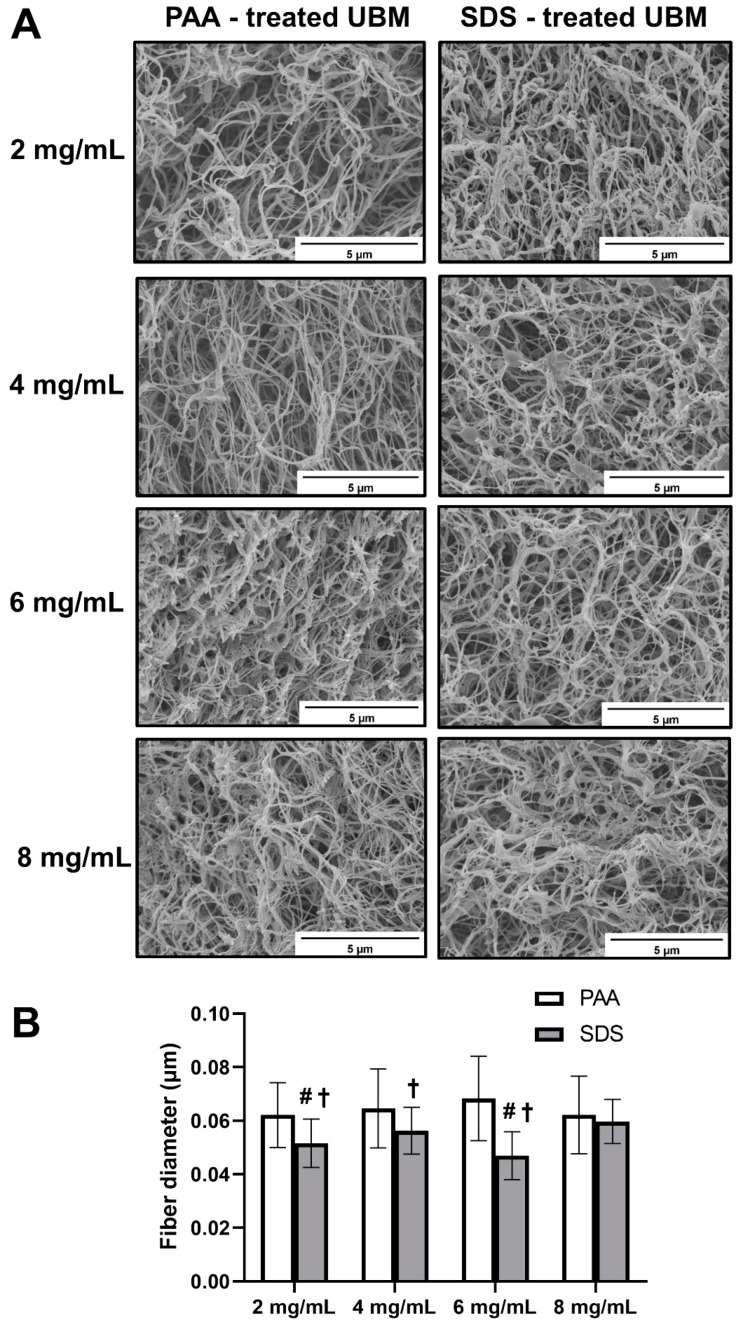
Hydrogel surface morphology of hydrogel from PAA and SDS groups. Scanning electron micrographs of UBM hydrogel surface (**A**) (scale bar 5 µm). Fiber size (**B**) of hydrogels. # denotes statistically difference from the 8 mg/mL concentration of the same decellularization group type, and † denotes statistically difference between PAA and SDS groups at the same concentration (*p* < 0.05).

**Figure 6 polymers-12-03007-f006:**
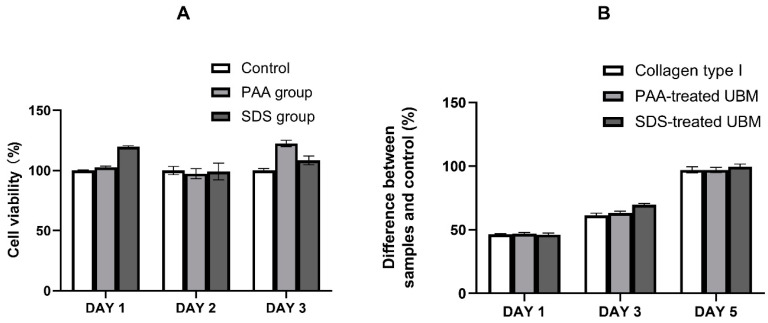
UBM hydrogel cytocompatibility. (**A**) In vitro cytotoxicity of the ECM powder studied by MTT assay. (**B**) Cell proliferation on hydrogel surface evaluated by Alamar blue assay.

**Figure 7 polymers-12-03007-f007:**
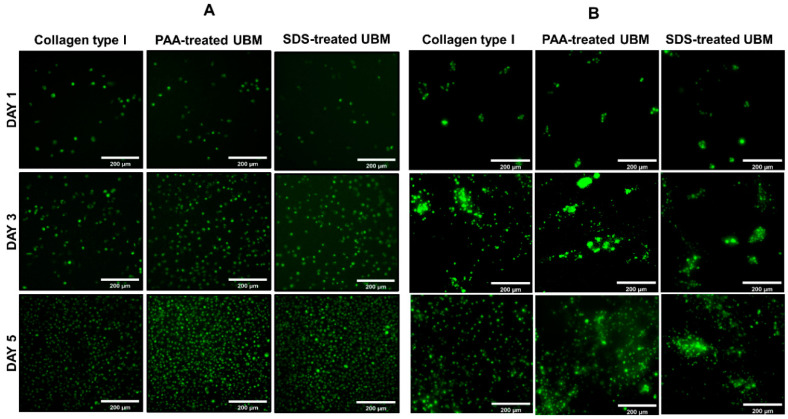
UBM hydrogel cytocompatibility. Live/dead assay of L929 cell cultured on hydrogel surface (**A**) and within hydrogel (**B**) (scale bar: 200µm).

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
