# Peer review of "Characterization of Porcine Urinary Bladder Matrix Hydrogels from Sodium Dodecyl Sulfate Decellularization Method"

_polymers, 2020, doi:10.3390/polym12123007_

Round 1

Reviewer 1 Report

In ‘Characterization of Porcine Urinary Bladder Matrix Hydrogels from Sodium Dodecyl Sulfate Decellularization Method’, Kao et al. compared the use of sodium dodecyl sulfate to peracetic acid for the decellularization of urinary bladder matrix. They characterized the decellularization efficiency, gelation kinetics, biochemical composition, surface morphology, rheological properties, and cytocompatibility.  The two methods produced similar hydrogels. The background information and novelty of the study are clearly presented.  The methods are succinctly but relatively clearly reported.  The discussion is also fine; however, several points within the results should be clarified.

Specific comments:

To clarify, is the second treatment in the decellularization protocol for both control and testing group performed with PAA (line 95). What is the rationale for this step?

Line 107: Is fluorescent images meant instead of fluorescein images?

What is the rationale for performing the shear viscosity measurements at 10C?

Section 2.9.2: What concentration of pre-gel solution was used?

Figure 1D: Are the scale bars correct?  If the H&E is showing intact nuclei, this size scale would indicate they are sub-micron in size, which does not match with the DAPI staining in Figure 1C.

Figure 3: It would be helpful to make the graphs in panels C-E slightly larger, or to use a larger font size.  In particular, it is difficult to read the symbols indicating the significant differences.

Line 241: Should this be Fig. 4A, B?

Figure 5: It would be helpful to add a more visible scale bar on the SEM images.

Section 3.6 and throughout: Cytocompatibility (or cytotoxicity) would be a better term to use than in vitro biocompatibility, as demonstration of true biocompatibility also involves in vivo evaluation.

Lines 270-271 and 276-277 and Figure 6: It seems that some of the cell culture results are missing. The results from the experiment where the cells are cultured on the surface of the hydrogels (Figure 6B) are not reported in the text.  Or alternatively, the caption for Figure 6B is incorrect.

Line 401: Was the collagen hydrogel concentration 2.5 mg/ml or 3 mg/ml as reported elsewhere?

Reviewer 2 Report

Authors presented SDS assisted UBM materials. The experiments are well designed and the manuscript is in good form. However, some questions must be solved  and some suggestions are listed. The paper might be published after minor revisions.

  1. I’m confused with some sentences. The manuscript should be polished.
  2. Scale bars in Fig. 1 and Fig. 5 are not clear.
  3. Lower error bars should be added in Fig. 1A, Fig. 2, etc.
  4. Space between numbers and units must be added.
  5. I don't think Triton is a suitable reagent used before SDS, please add some discussions.
  6. Please double-check the procedure of the in vitro cytotoxicity, did I miss any information about the control group?
  7. Optical photographs of the samples are suggested, maybe in SI.
